# Clinical Perspectives of Non-Coding RNA in Oral Inflammatory Diseases and Neuropathic Pain: A Narrative Review

**DOI:** 10.3390/ijms23158278

**Published:** 2022-07-27

**Authors:** Jelena Roganović, Nina Petrović

**Affiliations:** 1Department of Pharmacology in Dentistry, School of Dental Medicine, University of Belgrade, 11000 Belgrade, Serbia; 2Department of Radiobiology and Molecular Genetics, Vinča Institute of Nuclear Sciences, National Institute of the Republic of Serbia, University of Belgrade, 11000 Belgrade, Serbia; dragoninspiration@yahoo.com; 3Institute for Oncology and Radiology of Serbia, 11000 Belgrade, Serbia

**Keywords:** non-coding RNA, oral inflammatory diseases, precision medicine, neuropathic pain, salivary diagnostics, regenerative dentistry

## Abstract

Non-coding RNAs (ncRNAs) represent a research hotspot by playing a key role in epigenetic and transcriptional regulation of diverse biological functions and due to their involvement in different diseases, including oral inflammatory diseases. Based on ncRNAs’ suitability for salivary biomarkers and their involvement in neuropathic pain and tissue regeneration signaling pathways, the present narrative review aims to highlight the potential clinical applications of ncRNAs in oral inflammatory diseases, with an emphasis on salivary diagnostics, regenerative dentistry, and precision medicine for neuropathic orofacial pain.

## 1. Introduction

Oral inflammatory diseases are commonly observed pathologies in everyday dental clinical practice and via activation of the systemic immune response can lead to the development of systemic inflammatory disorders and impairment of the individual’s general health [1]. In order to improve clinical outcomes and increase patient quality of life, there is a tendency to incorporate molecular profiling into clinical decision-making and molecular signature-guided therapies. Based on the understanding of non-coding RNAs (ncRNAs) in the regulation of inflammatory signaling, new avenues for ncRNA in diagnostics and therapeutic intervention in inflammatory diseases have opened.

## 2. Non-Coding RNA Transcriptome

It has been proposed that approximately 70% of the human genome is transcribed into mRNA, but only around 2% are protein-coding [2,3,4], suggesting that a very tiny proportion of the human genome sequences and elements translate into proteins. According to this, a vast majority of transcribed sequences of RNA molecules are regulatory elements. Non-coding RNAs are genome elements that do not encode for amino acids [4]. They are involved in various cellular processes and interfere with signaling pathways. Non-coding RNAs are localized in almost every cellular compartment and interact with nucleic acids and proteins, thus changing their conformation and functions. Some non-coding RNAs interact with chromatin, contributing to chromatin remodeling, and thus shaping gene activity. In the backbone of every pathological condition lies non-coding RNA transcriptome changes, and their interactome. The localization and abundance of non-coding RNA are tissue-specific [5] and can change in response to a wide span of stressogenic factors and cellular and environmental events. Non-coding RNA transcriptome changes should be more closely discovered in association with disease development and have the potential to be utilized as diagnostic, predictive, and prognostic biomarkers; parameters for the stratification of patients into more specific groups; and potential targets for therapeutics. The non-coding RNA transcriptome comprises RNA molecules of various structures, lengths, and functions [4]. The basic classification on long non-coding (lncRNA) and small non-coding RNA is based on their length.

Transfer RNA (tRNAs) and ribosomal rRNAs (rRNAs) were among the first RNAs found that do not translate into proteins. However, high-throughput “whole transcriptome” techniques discovered seemingly insignificant but now very important regulators of all cellular processes in the two additional classes of non-coding RNAs: long non-coding and small non-coding RNA molecules. [6]. Classification of non-coding RNAs is based mostly on their length, but there are other types of non-coding RNA based on their special features such as packing into microvesicles, including extracellular RNAs [7].

### 2.1. Long Non-Coding RNA

LncRNAs are endogenous cellular RNAs lacking an ORF (open reading frame) or with a shorter ORF than mRNA, which is used as one of the criteria for distinguishing lncRNAs from messenger RNAs [8,9]. LncRNAs are classified as more than 200-nucleotide-long RNA molecules [10]. There are more than 16,000 lncRNA genes, while estimations suggest more than 100,000 lncRNAs [11]. The expression and abundance of lncRNAs is tissue specific; they have specific cellular localization, and have the ability to alter signaling pathways by interacting with nucleic acids and proteins. They are involved in gene expression regulation at the pre-, post-, and transcriptional level via changes in the chromatin state, and transcriptional activity, splicing, and translation [11,12]. LncRNAs can be subclassified into the groups of very long intergenic RNAs longer than 10 kb, and macroRNAs, which are pathway specific [13]. Five additional subclasses of RNA transcripts can be distinguished: intronic, intergenic, sense, antisense, and bidirectional [8]. Regarding the functional significance of lncRNA, one should be cautious, and conclusions should rely on specific studies with genetic quantitative loss-of-function or gain-of-function models in each species due to the different levels of lncRNA evolutionary conservation [9]. Namely, it has been suggested that lncRNA conservation between species could be considered at four dimensions: at the sequence level, where lncRNAs from different species can have sequence homology and thus similar transcripts; at the structural level, where similar structures could be produced despite a lack of lncRNA sequence homology; at the functional level, where similar functions could be executed despite lncRNAs’ different sequences and structures; and at the transcriptional level, where the locus of transcription is conserved, thus mediating functions despite different lncRNA transcripts [9].

### 2.2. Small Non-Coding RNA

Small non-coding RNAs (sncRNAs) are 20–200-nucleotide (some around even 400 nts in length) RNA molecules involved in the regulation of gene expression, transcription, translation, splicing, RNA modification, and methylation by the interaction with target mRNA, either through full or partial complementarity [14]. Different types of snRNAs such as microRNAs (miRs), endogenous short interfering RNA (endo-siRNA), piwi-interacting RNA (pi-RNAs), transfer RNA (tRNAs), transfer RNA fragments (tRFs), small nuclear (snRNAs), small nucleolar (snoRNAs), small Cajal-body RNAs (scaRNAs), YRNAs (small stem-loop RNA structures), and short hairpin RNA (shRNA) are synthesized through various biogenesis and maturation pathways by various enzymes and by interaction with different proteins, such as Argonaute or Pi-wi like, combined into the ribonucleoproteins [14,15,16]. Ribonucleoprotein complexes guide non-coding RNA to the target and enable them to interact not only with RNA molecules but with proteins and DNA in some cases, such as microRNAs, thus forming a very complex coding-non-coding RNA-DNA-protein reactome [17,18,19].

So far, the vast number of studies have focused on microRNAs as critical molecules governing development and stress responses. Very recent studies have paid attention to other sncRNAs, tRNA and tRFs, that accumulate in the stressed cell [20]. Because both sncRNAs have a Dicer-dependent biogenesis pathway, it has been proposed that tRFs could function akin to an miR to inhibit the translation or cleave partial complementary target sites [21]. The canonical pathway of miR biogenesis requires two enzymes: Drosha and Dicer. Primary miR transcripts are first cleaved by the nuclear “microprocessor” complex, which contains the enzyme Drosha, and then exported to the cytoplasm, where they are further processed by another enzyme, Dicer, forming the mature miR. Dicer is critical for most miRNAs, but the 5p miRNAs appear to be produced to some extent even without Dicer. Moreover, unlike canonical miR-21-5p, the synthesis of noncanonical miRs, including miR-320a-3p and miR-484–3p, is Drosha independent [22].

In addition to those described above, it is noteworthy to mention extracellular RNAs (exRNAs) [7] and competing endogenous RNAs (ceRNAs) packed into lipid or protein particles, such as exosomes, microvesicles, or oncosomes, which are especially significant for cancer research, which bare and transfer very important genetic and biological information between cells and organ systems [7]. The competing RNA (ceRNA) term is associated more with their function and role rather than the length or another structural feature or localization. The competing RNA phenomenon is associated more with interaction with long and small non-coding RNA and other transcripts, thus acting as sponges by controlling the amounts of active (free to interact with) non-coding RNA transcripts [23].

### 2.3. Challenges Related to the Clinical Application of ncRNAs

The introduction of single-cell sequencing has enabled investigation of not only the landscape of ncRNAs but also their cellular function. Furthermore, bioinformatic tools are becoming more sophisticated and are used for analyzing ncRNA sequencing data. However, detection of ncRNAs still remains challenging due to the low expression and unique features of certain ncRNAs and the bias of RNA sequencing and bioinformatic methods, leading to erroneous identification of ncRNA species [24].

Currently, eleven RNA-based therapeutics have been approved by the US Food and drug administration and/or the European Medicines Agency involving antisense oligonucleotides (ASOs) or small interfering RNAs (siRNAs) while others, including miRNA mimics and antimiRs, are in phase II or III clinical investigations [25].

The use of miR-based therapeutics has several advantages [26]. Namely, miRs are naturally occurring molecules in human cells, contrary to synthetic chemotherapy compounds or ASOs, and all the mechanisms in cells for their processing and downstream gene target selection are available. Additionally, miRs, by targeting multiple genes within one pathway, show a broader yet specific response [26]. It is noteworthy that circulating miRs (in serum, saliva, and other body fluids), due to their stability and distinctive function, have the advantage of being biomarkers compared with other biomarkers such as proteins (cytokines). Moreover, as controllers of gene transcription, miRNAs and their expression have a higher probability of being related to clinical variables and since they reflect cellular disturbance that occurs years before the appearance of related clinical signs, they enable disease-preventing actions [27].

Clinical applications of all RNA-based therapeutics are hindered by several issues: specificity, related to undesired effects due to uptake in non-target cells or overdosing; delivery, mainly related to inefficient intracellular delivery and the lack of suitable delivery vehicles; and tolerability, caused by the recognition of RNA molecules by pathogen-associated molecular pattern (PAMP) receptors, causing strong immune effects [25]. Therefore, in order to achieve the application of ncRNA as therapeutics and biomarkers, further research should focus on immunogenicity screening, extensive pharmacodynamic and pharmacokinetic studies regarding delivery systems, and chemical modifications to improve specificity.

## 3. ncRNAs and Oral Inflammatory Diseases

Every disease is a consequence of the specific interactions among the genetic and epigenetic backgrounds of each individual and cellular and environmental conditions. During the last decade, it has been discovered that the non-coding elements of the genome underlie the onset and progression of inflammatory diseases [12].

Pulpitis is a chronic inflammatory condition of the dental pulp, mostly caused by cavities [28]. Periodontal disease is a result of the complex interaction among inflammatory and immune responses induced by pathogenic bacteria [29] and the individual’s genetic and epigenetic background. Periodontitis is mainly caused by bacterial infection affecting periodontal tissue, which can lead to the loss of alveolar bone attachment [30]. Peri-implantitis is an inflammatory process of the tissues around an osseointegrated implant and may result in supporting bone loss, and miRs, such as ncRNAs, could be engaged in the prognosis of peri-implant bone resorption [31].

Sjögren’s syndrome (SS) is a rheumatoid pathological condition predominantly affecting the salivary and lachrymal glands, leading to oral dryness and salivary gland swelling. Patients with SS show an increased risk of developing non-Hodgkin’s lymphoma [32]. Oral lichen planus (OLP) is a premalignant epithelial oral lesion, with the potential to malignantly transform [33]. Oral squamous cell carcinoma (OSCC) is a malignant epithelial tumor with low overall survival rates and poor prognosis, regardless of the advances in surgery, chemotherapy, and radiotherapy [34]. Growing evidence supports an association between oral squamous cell carcinoma (OSCC) and chronic inflammation and the involvement of long non-coding transcriptome alterations [34].

### 3.1. Long Non-Coding RNA and Oral Inflammatory Diseases, Premalignant States, and Oral Squamous Cell Carcinoma

LncRNAs were shown to be closely associated with oral inflammatory diseases and malignant transformation in the oral cavity epithelium, but the mechanisms underlying these pathological conditions are still unknown. Long non-coding RNAs were shown to be immunomodulatory [35]. Thus, antisense non-coding RNA in the INK4 locus (*ANRIL*) regulates the STAT1 pathway and thus the production of the proinflammatory cytokine interferon-gamma (IFN-γ). ANRIL is one of the regulators and a component of the NF-kB pathway, and after TNFα treatment, it induces IL-6 and IL-8 expression [36], indicating that it is an important regulator of inflammation underlying the pathology of this type of disease. Upregulation of another proinflammatory long non-coding RNA, of lncRNA MEG3 (lncRNA maternally expressed gene 3), was associated with pulpitis progression while its downregulation was associated with dental pulp regeneration [37]. The authors also found that inhibition of lncMEG3 lowered the secretion of proinflammatory cytokines in dental pulp cells treated with LPS, probably via p38/MAPK signaling pathway regulation [37]. LncRNA FGD5 antisense RNA1 was shown to be underexpressed in gingivae in patients with periodontitis by blocking miR-142 to silence NF-kB signaling and the inflammatory response [38]. LncRNA MALAT1 acts as a sponge to miR-20a and increases inflammatory processes in periodontal tissue via activation of the toll-like receptor 4 pathway [39]. LncRNA DQ786243 was shown to be overexpressed in the CD4+ lymphocytes of oral lichen planus patients compared with healthy individuals and associated with increased proinflammatory miR-146a through Foxp3 activation, which downregulates the NF-κB pathway and, in turn, affected the resulting LncRNA DQ786243 expression [40]. It has been shown that lncRNA TMEVPG1 was increased in CD4+ T helper cells in patients with Sjögren syndrome compared with healthy individuals [41], indicating its potential involvement in the development of SS. It was reviewed by Benedittis et al. that lncs LINC00426, NRIR, CYTOR, TPTEP1, BISPR, AC017002.1, n336161, LINC00426-003, NR_002712, LINC02384, TCONS_l2_00014794, lnc-UTS2D-1:1, and n340599 expression was altered in PBMCs or salivary glands in SS patients [42].

Jia et al. [43] showed the potential of four lncRNAs (*ENST00000412740*, *NR_131012*, *ENST00000588803,* and *NR_038323*) to distinguish early stage from advanced-stage OSCC. They also showed significant differences between OSCC and healthy controls, indicating that these four lncRNAs not only have prognostic potential but diagnostic potential as well [43]. Other lncRNAs frequently associated with oral cancers are lncRNA cancer susceptibility candidate 9 (lncRNA CASC9) [44] and lncRNA HOTAIR, which was described as an oncogene that boosts the invasive and metastatic potential of OSCC [45]. LncRNA CASC9 was overexpressed in OSCC tissue and SCC15, TSCCA, and CAL27 oral cancer cell lines compared with healthy matched tumorous tissue and HOMEC, a normal cell line derived from oral keratinocytes. Higher lncRNA CASC9 levels were also associated with an advanced T stage, positive lymph node status, and an advanced clinical stage [44]. LncRNA HOTAIR showed higher expression levels in OSCC and cell lines and advanced TNM stages, higher tumor grade, and positive lymph node status compared with healthy oral mucosa epithelium and a normal cell line derived from keratinocytes from the oral cavity. Furthermore, LncRNA HOTAIR acts as a sponge for miR-326, thus decreasing its ability to silence the translation of oncogenes, such as metastasis-associated gene 2, which was confirmed by Tao et al. [45].

### 3.2. Small Non-Coding and Oral Inflammatory Diseases, Premalignant States, and Oral Squamous Cell Carcinomas

In oral diseases and pathology, microRNAs are the most investigated small non-coding RNA with the highest potential to be utilized in clinical practice. MicroRNAs play a very important role in dental pulp pathology via their role in regulating the immune response and inflammation [46]. miR-21 was shown to mitigate the inflammatory signaling in LPS-stimulated dental pulp cells [47]. Bacterial infection with *P. gingivalis* lipopolysaccharides (LPS) was shown to induce miR-584 overexpression, which induced IL-8 production and inflammation in the gingival epithelium [48,49].

Kamal et al. [50] compared the differential expression of miRs from the saliva and plasma of patients with chronic periodontitis (CP) with miRs extracted from healthy controls. The authors pointed out that the arrays from saliva and plasma differed from each other and found that miR-let-7d/miR-103a-3p/126-3p/150-5p/199a-3p/4485-5p/6088/6821-5p were significantly lower in the saliva and plasma of the CP patients compared with the controls, indicating that these miRs might be used as diagnostic and prognostic factors of CP [50]. MicroRNA miR-146a and miR-155 deserve special attention because it has been shown that these miRs can modulate immune responses [51]. Their upregulation was observed in the crevicular fluid of patients with chronic periodontitis associated with diabetes mellitus type II [52]. Sipert et al. [51] investigated the differences in the miR expression levels in cultivated fibroblasts in dental pulp, gingival, and periodontal ligament fibroblasts from the same individual by microarray and RT-qPCR analysis. The authors showed a cell-type-specific miR expression pattern and an increase in proinflammatory miR-146a in gingival fibroblasts after stimulation with LPS, and that miR-155 in gingival fibroblasts was decreased after LPS addition [51]. By employing differential expression analysis, miR-517/525/624/3128/3658/3692/3912/3920/4683/4690 were identified as predictors of periimplantitis in the five years after implant surgery, which regulate critical processes in peri-implant tissues, such as inflammation or cellular proliferation [27].

Exosomes represent cellular particles that contain proteins, lipids, coding and non-coding RNAs, microRNA, and cytokines, thus having the ability to transfer information among neighboring and distant cells. It has been also shown that exosomal microRNA can modulate the immune response and inflammation [53,54]. Zheng et al. [55] showed that dental pulp stem cell-derived small extracellular vesicles (DPSCs-sEV) and their cargo, including 81 miRs, have immunomodulatory features in dental pulp cells. Especially, miR-125a-3p jumped was notable for its immunomodulatory potential by regulating the NF-κΒ and toll-like receptor (TLR) axis via silencing of the inhibitor of nuclear factor-kappa B kinase subunit beta (IKBKB) [55]. Exosome-derived microRNA molecules can alter the translation of mRNAs in immune cells [54]. MicroRNA miR-142-3p, which was found in exosomes from T cells, was associated with exocrine gland malfunction in SS [56]. Some other miRs from exosomes such as miR-124a/192-5p and miR-150-5p were associated with an impaired immune response in rheumatoid arthritis [54].

In OLP, miRs are the most studied small non-coding RNA. Gassling et al. [57] identified 16 altered miRNAs by microarray analysis in 7 patients with OLP. Oncogenic miRs, such as miR-21, were shown to be significantly upregulated, meaning that OLP might be a precursor state for malignant transformation. Additionally, miR-31/132/143/155/15a/342-3p were also differentially expressed in OLP compared with healthy controls [57]. Scapoli et al. [58] found that miR-21/23/25/146b/489/129/338/212, among others, were overexpressed in OSCC and that tumor-suppressive miR-34a/520h/197/378/135b, and miR-224 were lower than in healthy control samples. Furthermore, miR-let-7i, miR-155, and miR-146a downregulation was associated with OSCC metastasis [58].

A study investigated circulating small non-coding RNA derived from seven male patients with oral cancer and found a significant difference in the distribution of small non-coding RNA between the cancer and healthy control groups [59]. According to their results, a vast majority of investigated small non-coding RNA (50%) were miRs (miR-103-3p and miR-107 emerged as the most important, and associated with the tumor size), 38% were YRNAs, and 10% were tRNAs while snRNAs, rRNA, snRNA, and snoRNA together contributed only 1% of the small-noncoding RNA transcriptome [59]. These findings indicate the importance of microRNA in future clinical practice for oral disease diagnosis, prognosis, and treatment. The list of lncRNA and sncRNA associated with oral inflammatory diseases was presented in Table 1.

## 4. Clinical Perspectives of ncRNAs in Oral Inflammatory Diseases

Due to their involvement in DNA translational control, their regulation of mRNA and protein expression levels, and their ability to reprogram cellular signaling pathways in oral inflammatory diseases, ncRNAs could be used to diagnose and predict disease and to improve patient-tailored treatments as an integral part of precision medicine for oral inflammatory diseases (Figure 1).

### 4.1. ncRNAs as Salivary Diagnostic Markers of Oral Inflammatory Diseases

Saliva sampling represents a cost-effective and non-invasive procedure while ncRNAs, due to their short size, body fluid stability, and main location inside exosomes, represent very suitable salivary biomarkers. Prominently investigated, microRNAs are engaged in the regulation of cytokine expression and have been established as significant in the pathogenesis of oral inflammatory diseases. As a result, their evaluation in body fluids may be helpful in assessing disease status and progression and in the evaluation of the treatment process.

Salivary miR-21 can be used as a diagnostic marker for oral potentially malignant disorders, showing a specificity of 66% and sensitivity of 69% and an area under the receiver operating characteristic (ROC) curve (AUC) of 0.82 [60]. Regarding OSCC, ROC curve analysis of salivary miR-424, miR-31, and miR-345 showed that each miR had limited power individually to differentiate between OSCC and healthy controls, with miR-345 having the largest AUC of 0.77. However, their combination could differentiate well between OSCC and control samples, with an AUC of 0.87, specificity of 0.77, and sensitivity of 0.86 [61]. Furthermore, the ROC analysis performed by He et al. [62] showed that salivary exosomal miR-24-3p has diagnostic accuracy for OSCC, with an AUC of 0.74, while miR-512-3p and miR-412-3p, with AUC values of 0.85 and 0.87, were also reported as perspective diagnostic markers for OSCC [63].

Patients with periodontitis show higher expression of miR-146a/155 in crevicular fluid, showing high diagnostic accuracy for periodontitis, with an AUC >0.9. [52]. In saliva, the AUC, specificity, and sensitivity of salivary miR-155 in diagnosing periodontitis were 0.88, 78%, and 97.14%, respectively, and those of miR-146a in diagnosing PD were 0.75, 58.54%, and 88.57%, respectively [64]. The miR expression profile in saliva may discriminate patients with primary Sjögren’s syndrome from those with Sjögren-like disease. Namely, analysis of the salivary miRs revealed that the AUC for miR-17-5p was 0.87, let-7i-5p was 0.91, and miR-328-3p was 0.84 when used as single biomarkers. Furthermore, the combination of miR-17-5p and let-7i-5p showed an AUC of 0.97 and, similarly, when all three miRNAs were combined while the four miR-17 family members (i.e., miR-17-5p/106a/106b/20b-5p) in combination yielded an AUC of 0.95 [65]. The list of salivary miRs with diagnostic value for oral inflammatory diseases is presented in the Table 2.

We performed bioinformatic analysis (miRnet) in an attempt to identify the shared target genes of salivary microRNAs with diagnostic value for oral inflammatory diseases. miRNet is an online network tool that visualizes the interactions between miRs and their targets [66]. According to miRNet, we discovered the shared targets of 17 miR molecules. One gene transcript (*PTEN*) was the potential target of 12 miRs (degree 12) and 3 gene transcripts (*CDKN1A*, *NFAT5*, and *KIAA1551*) were associated with 10 miRs (degree 10) (Table 3). The three miRs, miR-106a/106b-5p, and miR-20b-5p, interact with all four listed genes. Literature analysis of these four shared genes: CDKN1, PTEN, NFAT5, and KIAA1551 (RESF1), revealed that all genes are significant for immunological responses, mainly via regulation of the T cell responses. Namely, the CDKN1 gene encodes a potent cyclin-dependent kinase inhibitor p21, which has been shown to control autoimmune T cell autoreactivity without affecting normal T cell responses [67]. PTEN gene expression is significantly positively correlated with CD4/CD8A gene expression and T cell infiltration, especially T helper cells, central memory T cells, and effector memory T cells, in multiple tumor types [68] while PTEN loss predicts a poor therapeutic response and worse survival outcomes in patients receiving immunotherapy. NFAT5 plays a role in the development and activation of immune cells, especially T cells and macrophages contributing to autoimmune and inflammatory diseases [69]. KIAA1551 (RESF1, C12orf35) is highly expressed in the thymus, spleen, bone marrow, and liver, organs associated with the immune system and involved in T cell immunology and transplants [70]. Noteworthy, in silico analysis revealed that upregulation of salivary miR-146a/155 is predicted to upregulate ACE2 expression and essential SARS-CoV-2 receptors, and modulate the host antiviral response; thus, it could be related to the susceptibility of these patients to SARS-CoV-2 infection [71]. Evaluation of PTEN immunoexpression in human oral mucosa specimens showed that alteration of PTEN mediates oral submucous fibrosis pathogenesis and oral carcinogenesis [72]. Likewise, a cell culture study revealed that the NFAT5 transcription factor is able to promote oral cancer cell proliferation via changes in the subcellular localization of the epidermal growth factor receptor [73].

### 4.2. ncRNAs in Regenerative Medicine in the Field of Oral Inflammatory Diseases

As important players in the processes of differentiation and proliferation of stem cells, ncRNAs may represent an option for regenerative treatment (Figure 2). There is a growing body of evidence on the role of ncRNAs in human periodontal ligament stem cells (PDLSCs). Periodontal ligament is a highly specialized connective tissue that surrounds the tooth root, which contains mesenchymal stem cells capable of differentiating into osteoblasts, cementoblasts, and adipocytes; thus, it is considered to be a highly promising stem cell population for alveolar bone repair and regeneration in periodontal disease [74]. Substantial evidence has demonstrated that some lncRNAs, including MEG3, H19, and lncRNA-ANCR, may guide osteogenic differentiation of stem cells under physiological and pathological conditions [75]. In the study of Hao et al. [74], microarray analysis identified an miRNA profile of human PDLSCs that induced osteogenic differentiation. A total of 116 miRNAs were found to be differentially regulated, and the expression of 6 of them was validated: miR-654-3p/4288/34c-5p were found to be upregulated while miR-218-5p/663a/874-3p were downregulated during osteogenesis. However, it seems that the miR regulatory role depends on the microenvironment conditions. For instance, miR-17 seems to inhibit osteogenic differentiation of healthy human PDLSCs, but it has a promoting effect when the cells originate from periodontitis patients or are cultured under inflammatory conditions [76,77].

To illustrate the significance of miRNA in osteogenesis, induction or inhibition of several miRs, including miR-31, miR-26a, and miR-21, for calvarial defect repair was evaluated in vivo [78,79,80,81].

Using miRnet, we investigated the miRs-genes network of selected osteogenic miRs: hsa-mir-654-3p/4288/34c-5p/218-5p/663a/874-3p/21-5p/26a/31-5p (Table 4), and the top three genes with the highest degree (number of miR–mRNA interactions) were found: CDK6, E2F2, and FOXO3, represent perspective targets of bone regenerative medicine since they are involved in bone cell survival, proliferation, differentiation, and angiogenesis [82,83,84]. Noteworthy, FOXO3 transcription factor was found to associate with chronic periapical inflammation in periapical lesion specimens via IL-1β release regulation [85]. Furthermore, an animal study showed that FOXO3a signaling promotion could improve the mandibular bone loss caused by 1,25 dihydroxy vitamin D deficiency [86].

Based on previous studies, the research on ncRNAs during odontogenic differentiation of dental tissue-derived stem cells has mainly focused on miRs. Xu et al. [87] showed that upregulated expression of miR-21 and expression of signal transducer and activator of transcription 3 (STAT3) are associated with increased odontogenic differentiation of human dental pulp stem cells (DPSCs), promoted by tumor necrosis factor-α. Huang et al. [88] showed that miR-223-3p is expressed at a higher level in inflamed pulp and that overexpression of miR-223-3p in DPSCs significantly increased the levels of markers of odontoblast differentiation: dentine sialophosphoprotein and dentine matrix protein 1. Sun et al. [89] showed that miR-140-5p enhanced the proliferation of human DPSCs but inhibited the differentiation of human DPSCs via regulation of the lipopolysaccharide/toll-like receptor 4 signaling pathway. Downregulation of miR-224-5p may promote DPSCs’ proliferation and migration [90]. miR-34a promotes odontogenic differentiation of human stem cells from the apical papilla (SCAPs), a significant perspective for regenerative endodontics [91]. In this line, LncRNA H19 was reported to lead to enhanced odontogenesis of SCAPs via the miR-141/ SPAG9 signaling pathway [92].

We investigated the shared targets of six selected odontogenic miRNA molecules-hsa-miR-223-3p/21-5p/34a/140-5p-141/224-5p, and the top four genes with the highest number of interactions with the investigated miRs are presented in Table 5. These genes represent widespread regulators of dental pulp stem cell processes, including: quiescence, proliferation, metabolism, differentiation and lineage choice, cell death and survival, self-renewal, and angio-/vasculogenesis [93,94,95,96]. Immunohistochemical data in humans shows that the expression of VEGF is strongly positive in the inflammatory infiltrate in irreversible pulpitis, reflecting the decrease in the microvessel density in irreversible pulpitis [97]. Both VEGF and IGF, by contributing to odontogenic differentiation of DPSCs, represent perspective bioactive molecules in dental pulp tissue engineering [98,99].

### 4.3. ncRNAs as Biomarkers and Perspective Therapeutics for Neuropathic and Inflammatory Pain

Chronic orofacial pain is usually caused by inflammation and tissue or nerve injury, but it continues even after the initial injury has healed. It is usually characterized by ongoing or intermittent burning pain, an enhanced response to noxious stimuli (hyperalgesia), or pain in response to normally innocuous stimuli (allodynia), accompanied by distress, fatigue, and depression. Current treatment for this disorder has had limited success and new therapeutic strategies, based on precision pain medicine, are warranted. In this regard, engagement of ncRNA in precision oral neuropathic pain medicine represents a potential strategy (Figure 3).

ncRNAs have been identified in pain-related regions in the human nervous system and, following nerve injury in humans, there are highly significant correlations between the abundance of miR-29a and miR-500a in human lingual nerve neuromas and the pain VAS score [100], which suggests a potential contribution of specific miRNAs to the development of chronic neuropathic pain. Lutz et al. [101] proposed a model for the mechanism by which microRNAs contribute to chronic inflammatory and neuropathic pain, and it includes engagement of inflammatory mediators. Namely, the increase in inflammatory mediators, such as interleukin (IL)-1β, after injury induces a change in the expression of miRs in dorsal root ganglion (DRG) neurons, resulting in an alteration of pain-related genes and an increase in DRG neuronal excitability and pain hypersensitivity (hyperalgesia and allodynia). Indeed, miR-146a/199a/558 were shown to be involved in pain-related pathophysiology of osteoarthritis, linked to the expression of cyclooxygenase-2 [102,103,104]. MicroRNA profiles could serve as blood biomarkers of neuropathic pain in humans. For instance, differential expression of 18 miRNAs was reported in blood from patients with complex regional pain syndrome [105]. Human miR-132-3p/146a/miR-21 were upregulated in the white blood cells of patients suffering from neuropathic pain [106,107]. Heyn et al. [108] found that in blood samples from neuropathic pain patients, upregulated miR-124a/155 were associated with reduced expression of Sirtuin 1 mRNA, leading to the development of neuropathic pain. On the other side, Liu et al. [109] found that downregulation of hsa-miR-101 expression in plasma from patients with neuropathic pain led to nuclear factor kappa B activation and consequent development of neuropathic pain.

Trigeminal neuralgia (TN), a common type of orofacial neuropathic pain, is characterized by severe, sudden pain in the trigeminal nerve distribution. In humans with TN, upregulation of circulatory miR-132-3p/146b-5p/155-5p/384 was observed compared to healthy controls while functional analysis indicated that miR-155-5p could directly target and downregulate nuclear factor-E2-related factor 2, which modulates the expression of inflammatory genes [110]. In animal models of TN, Xiong et al. [111] found that knockdown of lncRNA uc.48+ by siRNA could inhibit transduction of TN signals in rats. Li et al. [112] reported that lncRNA MRAK009713 expression was markedly increased in DRG in a rat model of TN, and downregulation of MRAK009713 significantly inhibited the nociceptive transmission and reduced both mechanical and thermal hyperalgesia. On the other side, lncRNAGm14461 expression was upregulated in the trigeminal ganglion in a mice model of TN and is associated with the pain transmission of TN via regulation of proinflammatory cytokines and CGRP expression [113].

Burning mouth syndrome (BMS) is a chronic pain condition characterized by burning sensation or pain felt in the oral mucosa. The etiology of BMS is multifactorial, including oral parafunctional habit, salivary gland dysfunction, or nerve injury, while menopausal disorders and diabetes may contribute to the severity. A recent study by Kim et al. [114] found that salivary exosomal miRNAs (miR-1273h-5p/1273a/1304-3p/4449/1285-3p/6802-5p/1268a/1273d/1273f/423-5p) were upregulated while 18 exosomal miRNAs (miR-27b-3p/16-5p/186-5p/142-3p/141-3p/150-5p/374a-5p/93-5p/29c-3p/29a-3p/148a-3p/22-3p/27a-3p/424-5p/19b-3p/99a-5p/548d-3p/19a-3p) were downregulated in BMS patients compared to controls, suggesting miRs could play an important role in the diagnosis and progression surveillance of BMS.

Patients with temporomandibular disorders (TMDs) frequently report pain deriving from either intra-articular or extra-articular structures. Differences in the perceived TMD pain between individuals make diagnosis and management of the TMD complex, requiring a personalized approach. A research study by Xu et al. [115] found that in synovial fibroblasts from patients suffering from osteoarthritis of the temporomandibular joint, eight miRNAs were upregulated and six miRNAs were downregulated, with miRNA221-3p being the most downregulated. The miRNA221-3p downregulation was attributed to an abundance of IL-1β (inflammation), and associated with induction of matrix metalloproteinases, MMP1 and MMP9, involved in joint injury. In another study, miR-140-5p was found to regulate temporomandibular joint osteoarthritis (TMJOA) via the TGF-β/Smad signaling pathway and might serve as a novel prognostic factor of TMJ degenerative changes [116]. A very recent study showed that the levels of miR-101a-3p were significantly lower in a rat inflammation model with TMJOA and involved in apoptosis of chondrocytes [117]. Using miR21 knockout mice, Zhang et al. [118], reported that miR21, via critical regulation of growth differentiation factor 5 in chondrocytes, regulates cartilage matrix degradation and contributes to the progression of TMJ-OA. A list of ncRNAs with potential as biomarkers of neuropathic orofacial pain is shown in Table 6.

## 5. Conclusions

Although still in its infancy, the implementation of precision medicine for oral inflammatory diseases is expected to have a significant impact on patient well-being. At the forefront of precision medicine (in dentistry) is the ability to identify unique characteristics in individual patients with oral inflammatory disease, allowing selection of a tailored treatment. Among the engaged methods, assessment of the in vivo molecular characterization (signature) of the disease and the host (mal)adaptive immune responses, regenerative medicine, and monitoring of drug and patient outcomes could rely on ncRNAs.

## Figures and Tables

**Figure 1 ijms-23-08278-f001:**
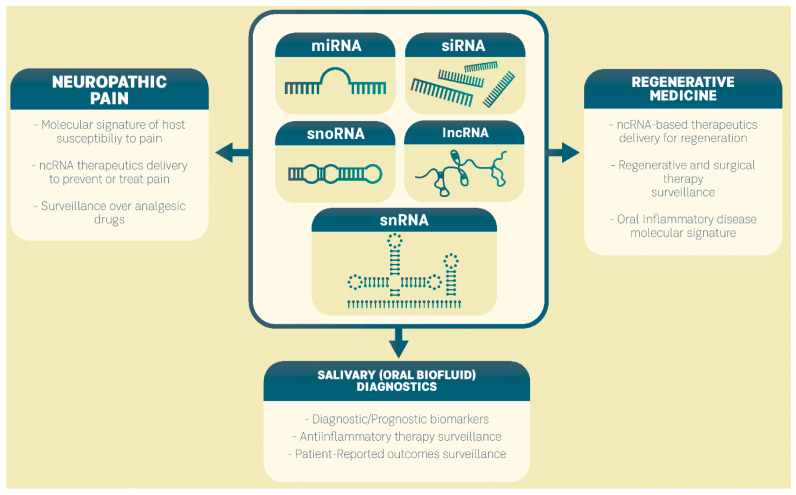
Clinical perspectives of ncRNAs in precision medicine of oral inflammatory diseases.

**Figure 2 ijms-23-08278-f002:**
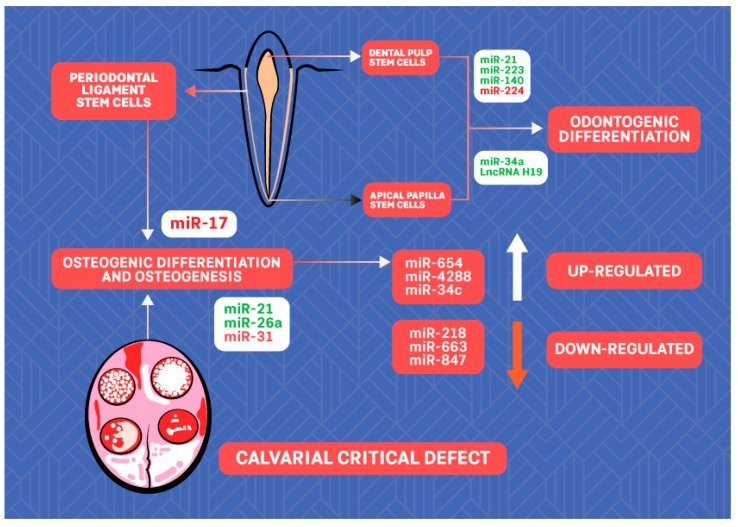
The scheme of ncRNAs with potential for regenerative dentistry. ncRNAs labeled in green promote while ncRNAs labeled in red inhibit osteogenic/odontogenic differentiation of stem cells.

**Figure 3 ijms-23-08278-f003:**
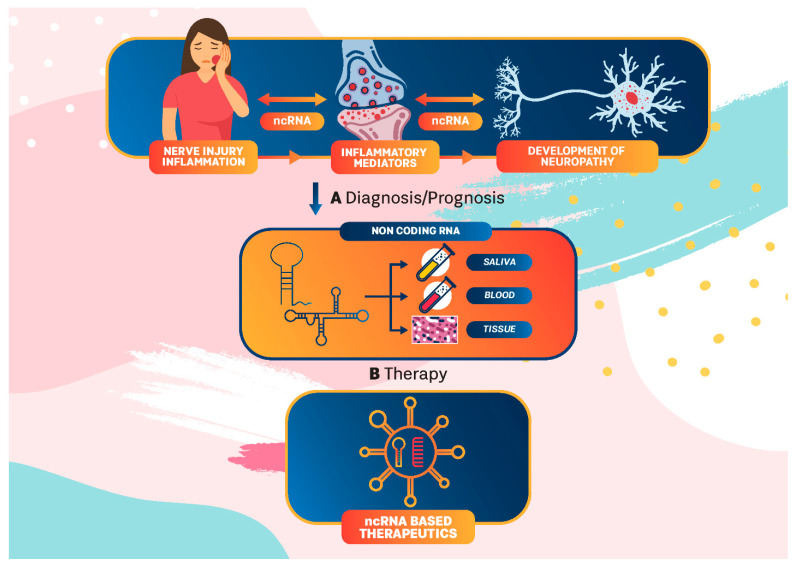
ncRNA in precision medicine for orofacial neuropathic pain. ncRNA interplay between injury/inflammation and neuropathy development could be used in diagnosis/prediction and in patient-tailored treatments as an integral part of precision medicine for neuropathic pain.

**Table 1 ijms-23-08278-t001:** LncRNA and sncRNA association with oral inflammatory diseases in clinical settings.

lncRNA or sncRNA	Association with Oral Inflammatory Disease	References
lncRNA MEG3	Upregulated in inflamed pulp. MEG3 downregulation inhibited the secretion of TNF-α, IL-1β, and IL-6 in LPS-treated hDPCs	[37]
lncRNA FGD5-AS1	Downregulated in gingival samples of periodontitis patients. Overexpression protects PDLCs via regulation of the miR-142-3p/SOCS6/NF-κB inflammatory signals	[38]
lncRNA TMEVPG1	Increase in CD4^+^ T cells in Sjogren syndrome patients	[39]
*ENST00000412740*, *NR_131012*, *ENST00000588803*, *NR_038323*	Downregulated in the plasma of patients with oral premalignant lesion and gradually increased with the malignant transformation process.	[43]
lncRNA CASC9	Upregulated in OSCC tissues; *CASC9* is strongly associated with tumor size, clinical stage, regional lymph node metastasis, and overall survival time in OSCC patients Enhances cell proliferation and suppresses autophagy-mediated cell apoptosis via the AKT/mTOR pathway	[44]
lncRNA HOTAIR	Upregulated in OSCC tissue Overexpression was positively correlated with TNM (tumor-node-metastases) stage, histological grade, and regional lymph node metastasis	[45]
miR-let-7d/miR-103a-3p/126-3p/150-5p/199a-3p/4485-5p/6088/6821-5p	Downregulated in both plasma- and salivary-exosomal samples of periodontitis patients	[50]
miR-517/525/624/3128/3658/3692/3912/3920/4683/4690	Predictors of peri-implantitis	[27]
miR-21/31/132/143/155/15a/342-3p	Upregulated in the oral mucosa of OLP patients	[57]
miR-21/23/25/146b/489/129/338/212	Overexpressed in OSCC tumors	[58]
miR-520h, miR-197, miR-378, miR-135b, miR-224, miR-34a	Underexpressed in OSCC tumors	[58]

**Table 2 ijms-23-08278-t002:** The list of salivary miRs with diagnostic value for oral inflammatory diseases.

Salivary miR	Oral Inflammatory Disease	Area under the Receiver Operating Characteristic Curve (AUC)	References
miR-21	Potentially malignant disorders	0.82	[60]
miR-424/31/345	OSCC	0.87	[61]
miR-24	OSCC	0.74	[62]
miR-512	OSCC	0.85	[63]
miR-412	OSCC	0.87	[63]
miR-155	Periodontitis	0.88	[64]
miR-146a	Periodontitis	0.75	[64]
miR-17/let7i	Sjögren syndrome	0.97	[65]
miR-17/106a/106b/20b	Sjögren syndrome	0.95	[65]

**Table 3 ijms-23-08278-t003:** Bioinformatic analysis (miRnet) of salivary miRs with diagnostic value for oral inflammatory diseases showing the top four target genes with the highest degree in the miR-mRNA regulatory network.

Target Gene	Degree	miR Involved in the Regulation of Target Genes
PTEN	12	miR-106b-3p/21-3p/486-5p/10b-5p/20b-5p/106b-5p/106a-5p/155-5p/17-5p/106a-3p/21-5p/155-3p
CDKN1A	10	miR-17-5p/106a-5p/106b-5p/20b-5p/10b-5p/345-5p/146a-5p/328-5p/512-5p/486-3p
NFAT5	10	miR-17-5p/155-5p/106a-5p/106b-5p/20b-5p/345-5p/146a-5p/31-5p/21-5p/24-3p
K1AA1551	10	miR-10b-5p/20b-5p/106b-5p/106a-5p/17-5p/106a-3p/21-5p/155-3p/27b-3p/512-3p

**Table 4 ijms-23-08278-t004:** Bioinformatic analysis (miRnet) of miRs involved in the regulation of osteogenic differentiation of human periodontal ligament stem cells and osteogenesis in vivo showing the top four target genes with the highest degree in the miR-mRNA regulatory network.

Target Gene	Degree	miRs Involved in the Regulation of Target Genes
CDK 6	4	miR-34c-5p/21-5p/26a-5p/218-5p
E2F2	4	miR-21-5p/26a-5p/218-5p/31-5p
FOXO3	4	miR-31-5p/26a-5p/21-5p/218-5p
NUFIP2	4	miR-218-5p/26a-5p/21-5p/874-3p

**Table 5 ijms-23-08278-t005:** Bioinformatic analysis (miRnet) of miRs involved in the regulation of odontogenic differentiation of human dental pulp and apical papilla stem cells showing the top four target genes with the highest degree in the miR-mRNA regulatory network.

Target Gene	Degree	miR Involved in the Regulation of Target Genes
IGF1R	6	miR-223-3p/21-5p/140-5p/34a-5p/141-3p/224-5p
CAPRIN1	4	miR-141-5p/223-3p/21-5p/34a-5p
E2F3	4	miR-140-5p/21-5p/34a-5p/141-3p
VEGFA	4	miR-141-5p/140-5p/21-5p/34a-5p

**Table 6 ijms-23-08278-t006:** ncRNAs as potential biomarkers of neuropathic orofacial pain.

Neuropathic Orofacial Pain	Biofluid or Tissue	Clinical Significance	References
Temporomandibular Disorders (TMDs)	Synovial fibroblasts and articular cartilage	Altered expression of miR221–3p/140-5p/101a-3p/21-5p was observed in degenerative TM joint disease and pain	[115]
			[116]
			[117]
			[118]
Burning Mouth Syndrome (BMS)	Saliva	There were upregulated exosomal miRNAs (miR1273h-5p/1273a/1304-3p/4449/1285-3p/6802-5p/1268a/1273d/1273f/423-5p) and downregulated miRNAs (miR-27b-3p/16-5p/186-5p/142-3p/141-3p/150-5p/374a-5p/93-5p/29c-3p/29a-3p/148a-3p/22-3p/27a-3p/424-5p/19b-3p/99a-5p/548d-3p/19a-3p) in BMS patients compared to healthy	[114]
Trigeminal neuralgia	Serum	In humans with TN, upregulation of circulatory miR-132-3p/146b-5p/155-5p/384 was observed compared to healthy controls	[110]
	Trigeminal ganglion	LncRNA Gm14461 promoted pain transmission in a mouse TN model	[113]
		LncRNA uc.48+ overexpression promoted pain transmission in a rat TN model	[111]
	Dorsal root ganglia	Downregulation of Lnc MRAK009713 reduced hyperalgesia in rats	[112]

## Data Availability

Not applicable.

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
