# Peer review of "Clinical Perspectives of Non-Coding RNA in Oral Inflammatory Diseases and Neuropathic Pain: A Narrative Review"

_ijms, 2022, doi:10.3390/ijms23158278_

Round 1
Reviewer 1 Report
The manuscript by Rognovic and Petrovic is a review of studies on the roles of non-coding RNAs in oral inflammatory diseases, neuropathic orofacial pain, and potential applications in regenerative dentistry. The review is a valuable overview of many studies that have largely focus on microRNAs (miRs) and long non-coding RNAs (lncRNAs). The authors also include an analysis of predicted miR targets using the miRNet resource to examine some of the possible regulatory roles. The authors conclude the review by advocating for increased research into additional diagnostic and therapeutic applications of non-coding RNAs in oral diseases.
While the review has utility, it overlooks two topics. First, the review does not describe conservation of non-coding RNAs across species. This is needed to provide context for mammalian studies that are summarized (e.g., lines 399-406, and 428-432). The lack of conservation also constrains studies of many non-coding RNAs. Second, the review does not describe how non-coding RNAs are currently being evaluated as clinical therapeutics. This would strengthen the authors argument that non-coding RNAs are viable therapeutics.
The authors’ analysis of miR targets is a major weakness of the review. The miRnet resource was used to find the miR targeting relationships shown in Figures 2, 3 and 4. However, each time they used the resource, they used a different degree filter parameter. The authors used a degree filter of 9 for Figure 2, a degree filter of 4 in Figure 3, and a degree filter of 3 in Figure 4. A reader could speculate that the authors were setting the degree filter parameter to limit the number of target genes. However, the rationale for setting the degree filter parameter is not clear. Furthermore, it is conceivable that a miR targeting a single gene could have important roles in a biological process. Lastly, the authors do not provide evidence that any of the predicted targets in the three figures were differentially expressed or functional in the diseases studied.
Additional tables that summarize the miRs and lncRNAs discussed in sections 3.1, 3.2, 4.1 and 4.2 would help the reader group the non-coding RNAs into categories.
The authors should consider briefly mentioning tRNA-derived fragments in their review of small RNAs in Section 2.2.
The authors should also consider mentioning that some small RNAs described in studies do not follow the canonical rules for miRs (e.g., miR-4485, miR-6088 and miR-6821).
The following are a list of important edits.
Line 33 - Substitute “vast” for “waste”.
Line 35 – Delete “, and thus do not direct protein synthesis”.
Line 38 – Insert “Some” at the beginning of the sentence that begins with “Non-coding RNA”.
Line 104 – Substitute “cavities” for “caries”.
Line 136 – Reword “sponging”.
Line 168 – Substitute “most” for “mostly”.
Lines 171-172 – Reword sentence.
Author Response
RESPONSES TO REVIEWER 1
Dear Sir/Madam,
Thank You so much for the comments and suggestions that have improved the manuscript significantly.
Comment: While the review has utility, it overlooks two topics. First, the review does not describe conservation of non-coding RNAs across species. This is needed to provide context for mammalian studies that are summarized (e.g., lines 399-406, and 428-432). The lack of conservation also constrains studies of many non-coding RNAs. Second, the review does not describe how non-coding RNAs are currently being evaluated as clinical therapeutics. This would strengthen the authors argument that non-coding RNAs are viable therapeutics.
Response: Accordingly, in 2.1. we added new sentences (highlighted in yellow), as well as new paragraph 2.3. Challenges related to the ncRNAs clinical application
Comment:The authors’ analysis of miR targets is a major weakness of the review. The miRnet resource was used to find the miR targeting relationships shown in Figures 2, 3 and 4. However, each time they used the resource, they used a different degree filter parameter. The authors used a degree filter of 9 for Figure 2, a degree filter of 4 in Figure 3, and a degree filter of 3 in Figure 4. A reader could speculate that the authors were setting the degree filter parameter to limit the number of target genes. However, the rationale for setting the degree filter parameter is not clear. Furthermore, it is conceivable that a miR targeting a single gene could have important roles in a biological process. Lastly, the authors do not provide evidence that any of the predicted targets in the three figures were differentially expressed or functional in the diseases studied.
Response: The different degree filter for visualization of different data set was used in order to each regulatory miR-mRNA regulatory network be readable. To avoid possible inconsistency and confusion, we decided to present data in tables instead of figures, presenting for each data set top 4 nodes (target genes) with the highest degree in miR-mRNA regulatory network. Results from studies on investigated genes have been presented, also (highlighted in yellow).
Comment: Additional tables that summarize the miRs and lncRNAs discussed in sections 3.1, 3.2, 4.1 and 4.2 would help the reader group the non-coding RNAs into categories.
Response: Additional tables that summarize miRs discussed in section 3.1, 3.2 and 4.1 have been added (Tables 1 and 2), while those discussed in 4.2 were presented in the Figure 2.
Comment: The authors should consider briefly mentioning tRNA-derived fragments in their review of small RNAs in Section 2.2.
The authors should also consider mentioning that some small RNAs described in studies do not follow the canonical rules for miRs (e.g., miR-4485, miR-6088 and miR-6821).
Response: Accordingly, we introduced new sentences in Section 2.2. highlighted in yellow.
Comment:The following are a list of important edits.
Line 33 - Substitute “vast” for “waste”.
Line 35 – Delete “, and thus do not direct protein synthesis”.
Line 38 – Insert “Some” at the beginning of the sentence that begins with “Non-coding RNA”.
Line 104 – Substitute “cavities” for “caries”.
Line 136 – Reword “sponging”. By blocking miR-142 to silence
Line 168 – Substitute “most” for “mostly”.
Lines 171-172 – Reword sentence.
Response: Accordingly, edits have been done (highlighted in yellow).
Hope we met all your suggestions correctly,
Sincerely, Jelena Roganović
Reviewer 2 Report
The manuscript reports a narrative review on non-coding RNAs in oral inflammatory diseases. The paper is of interest and in my opinion only a minor revision is necessary before publication.
Title: the design of the study should be clear reading the title. I would change the title in this way: “Clinical perspectives of non-coding RNA in oral inflammatory diseases: a narrative review”.
The same consideration applies to the Abstract: the authors should specify that this is a narrative review.
In the field of oral inflammatory diseases the authors missed to address peri-implantitis, which is a topic of growing interest given the wide use of dental implants in clinical practice. A paragraph on non-coding RNAs investigated in the diagnosis and prognosis of peri-implantitis should be added. i.e. see the following papers related to miRNAs:
- Sartori EM, das Neves AM, Magro-Filho O, Mendonça DBS, Krebsbach PH, Cooper LF, Mendonça G. The Role of MicroRNAs in the Osseointegration Process. Int J Oral Maxillofac Implant 2019; 34:397-410
- Menini M, Pesce P, Pera F, Baldi D, Pulliero A, Delucchi F, Izzotti A. MicroRNAs in peri-implant crevicular fluid can predict peri-implant bone resorption. Clinical trial with a 5-year follow-up. Int J Oral Maxillofac Implants 2021;36:1148-1157.
- Menini M, Dellepiane E, Pera F, Izzotti A, Baldi D, Delucchi F, Bagnasco F, Pesce P. MicroRNA in Implant Dentistry: from basic science to clinical application. MicroRNA 2021;10:14-28.
In order to address the stated aim of their study, the authors should better clarify the clinical perspectives of non-coding RNAs, and how they can be applied. What is already available for clinical use? What are the possible methods in order to analyse non-coding RNAs? What type of samples can be analysed (hard and soft tissue, fluids)? What are advantages and disadvantages of the different methods? What are the limits for clinical application at the moment? What else needs to be determined to ensure predictable efficacy and effectiveness of the described analyses? What should be addressed in future research?
The authors should explain why non-coding RNAs are in their opinion better biomarkers compared to other biomarkers i.e. cytokines. Advantages and disadvantages of non-coding RNAs should be clearly stated.
Since the present manuscript is focused on oral inflammatory diseases, the paragraph 4.3. “ncRNAs as biomarkers and perspective therapeutics for neuropathic and inflammatory pain” is not applicable, I suggest to delete it.
Author Response
Dear Sir/Madam,
Thank You so much for the comments and suggestions that have improved the manuscript significantly.
Comment: The manuscript reports a narrative review on non-coding RNAs in oral inflammatory diseases. The paper is of interest and in my opinion only a minor revision is necessary before publication. Title: the design of the study should be clear reading the title. I would change the title in this way: “Clinical perspectives of non-coding RNA in oral inflammatory diseases: a narrative review”. The same consideration applies to the Abstract: the authors should specify that this is a narrative review.
Response: Accordingly, we made changes in the Title and Abstract
Comment: In the field of oral inflammatory diseases the authors missed to address peri-implantitis, which is a topic of growing interest given the wide use of dental implants in clinical practice. A paragraph on non-coding RNAs investigated in the diagnosis and prognosis of peri-implantitis should be added. i.e. see the following papers related to miRNAs:
- Sartori EM, das Neves AM, Magro-Filho O, Mendonça DBS, Krebsbach PH, Cooper LF, Mendonça G. The Role of MicroRNAs in the Osseointegration Process. Int J Oral Maxillofac Implant 2019; 34:397-410
- Menini M, Pesce P, Pera F, Baldi D, Pulliero A, Delucchi F, Izzotti A. MicroRNAs in peri-implant crevicular fluid can predict peri-implant bone resorption. Clinical trial with a 5-year follow-up. Int J Oral Maxillofac Implants 2021;36:1148-1157.
- Menini M, Dellepiane E, Pera F, Izzotti A, Baldi D, Delucchi F, Bagnasco F, Pesce P. MicroRNA in Implant Dentistry: from basic science to clinical application. MicroRNA 2021;10:14-28.
Response: Accordingly, new sentences in sections: 3.0. 3.2 (highlighted in yellow) and Table 1, and new references were added
Comment: In order to address the stated aim of their study, the authors should better clarify the clinical perspectives of non-coding RNAs, and how they can be applied. What is already available for clinical use? What are the possible methods in order to analyse non-coding RNAs? What type of samples can be analysed (hard and soft tissue, fluids)? What are advantages and disadvantages of the different methods? What are the limits for clinical application at the moment? What else needs to be determined to ensure predictable efficacy and effectiveness of the described analyses? What should be addressed in future research? The authors should explain why non-coding RNAs are in their opinion better biomarkers compared to other biomarkers i.e. cytokines. Advantages and disadvantages of non-coding RNAs should be clearly stated.
Response: Accordingly, we added new subsection: 2.3. Challenges related to the ncRNAs clinical application
Since the present manuscript is focused on oral inflammatory diseases, the paragraph 4.3. “ncRNAs as biomarkers and perspective therapeutics for neuropathic and inflammatory pain” is not applicable, I suggest to delete it.
Response: As stated in the present section, the inflammation is closely related to neuropathic pain, e.g. chemical mediators, such as cytokines, chemokines, and lipid mediators, released during an inflammatory response have the undesired effect of sensitizing and stimulating nociceptors, their central synaptic targets or both, resulting in persistent neuropathic pain. (A. Ellis, D. L. H. Bennett, 2013). Therefore, we consider it is important to involve this section into the manuscript. However, if you consider it as not that close link, we suggest to add in title: Clinical perspectives of non-coding RNA in oral inflammatory diseases and neuropathic pain: a narrative review.
Hope we met your suggestions correctly,
Sincerely,
Jelena Roganović
Round 2
Reviewer 1 Report
The manuscript is vastly improved and the new tables are very helpful. My only suggestion is to replace "volcano plot analysis" with "differential expression analysis" in section 3.2.
Author Response
Dear Sir,
according to Reviewer's comment, edit has been done in section 3.2.